



# Interaction between cloud-radiation, atmospheric dynamics and thermodynamics based on observational data from GoAmazon 2014/15 and a Cloud Resolving Model

Layrson J. M. Gonçalves[1], Simone M. S. C. Coelho[1], Paulo Y. Kubota[1], Dayana C. Souza[1]

[1]National Institute for Space Research, Cachoeira Paulista, SP, 12630000, Brazil

*Correspondence to*: Layrson J. M. Gonçalves (layrsongoncalves@gmail.com)

**Abstract.** Observational meteorological data from the field experiment GoAmazon 2014/15 and data from numerical simulations with the Cloud-Resolving Model (CRM) called System for Atmospheric Modeling (SAM) are used to study the interaction between the cloudiness-radiation and the atmospheric dynamics and thermodynamics variables for a site located in the central Amazon region (-3.2° S, -60.6° W) during the wet and dry periods. The main aims are to (a) analyze the temporal series of the integrated cloud fraction, precipitation rate and downward shortwave flux; and (b) to determine the relationship between the integrated cloud fraction, radiative fluxes, and large-scale variable anomalies as a function of the previous day's average. The temporal series of the integrated cloud fraction, precipitation rate and downward shortwave flux from SAMS simulations showed physical consistency with the observations from GoAmazon 2014/15.  Shallow and deep convection clouds show to have meaningful impact on radiation fluxes in the Amazon region during wet and dry periods. Anomalies of large-scale variables (relative to the previous day's average) are physically associated with cloud formation, evolution and dissipation. SAM consistently simulated these results, where the cloud fraction vertical profile shows a pattern very close to the observed data (cloud type). Additionally, the integrated cloud fraction and large-scale variable anomalies, as a function of the previous day's average, have a good correlation. These results suggest that the memory of the large-scale dynamics from previous day can be used to estimate the clouds fraction. As well as the water content, which is a variable of the cloud itself. In general, the SAM satisfactorily simulated the interaction between cloud-radiation and dynamic and thermodynamic variables of the atmosphere during the periods of this study, being indicated to obtain atmospheric variables that are impossible to obtain in an observational way.

## 1 Introduction

The radiation and clouds interaction plays an essential role in the Earth's atmosphere because it directly affects the diurnal cycle of meteorological variables and convective processes (Yang and Slingo, 2001). The physical processes associated with cloud formation produce disturbances in the atmosphere that interact with several waves that propagate from the tropical region of the Pacific, influencing the active and passive convection cycles in remote regions. The main waves operating in the tropical region are the Rossby, Kelvin, and inertial gravity waves that span periods from a few days to several weeks



(Mather, 2005; Matsuno, 1966). For this reason, the weather and climate scales are strongly modulated by the clouds influencing the Earth's system energy balance (L'Ecuyer et al., 2019).

Simulating the formation, properties, and feedbacks of different clouds types is one of the greatest scientific community challenges (Bodas-Salcedo et al., 2014; Calisto et al., 2014; Su et al., 2010; Zhang, 2005). Previous works indicate that the

cloud simulation deficiency and the misrepresentation of cloud microphysical and macrophysical properties can generate inaccurate feedbacks in Global Climate Models (GCMs). Therefore, cloud-radiation feedback is the major source of uncertainty in weather and climate forecasting and climate change scenarios (Del Genio, 2012; Stephen A. Klein and Anthony Del Genio, 2006).

Clouds also play an important role in planetary albedo, reflecting the solar radiative flux to space, thus cooling the planet (Wielicki et al., 1995). Additionally, clouds act as a barrier to longwave radiation emitted by oceans and continents, absorbing and emitting upward and downward to the surface. This latter process intensifies the planet's greenhouse effect by warming (Chen et al., 2000; Patnaude and Diao, 2020).

The effects of clouds on the warming/cooling climate system will depend on several factors, such as the height of the base and top cloud, cloud fraction, optical properties, liquid or ice phase of the cloud particle (Liou, 2002; Wang et al., 2019). These properties will define the absorption, transmittance, and scattering properties of long and shortwave radiation (Hossein Mardi et al., 2019; Maghrabi et al., 2019).

Thus, understanding the processes that involve cloud formation and cloud interactions with radiation fluxes (called the cloud-radiation feedback process) is one of the main challenges for numerical modeling of the atmosphere, due to the characteristics and diversity of clouds types in the Earth system (Chen et al., 2000; Giangrande et al., 2017).

The knowledge obtained from observed data for the different physical processes related to clouds life cycle (i.e., generation,

development, and dissipation) and the cloud-radiation interactions is necessary to determine other not directly perceptible phenomena. Thus, new theories can be formulated, helping to improve the existing physical parameterizations in atmospheric circulation models. In this regard, several campaigns of data collection experiments were carried out in different regions, for example, GoAmazon 2014/15 (Giangrande et al., 2017), DYNAMO (Feng et al., 2014; Fliegel and Schumacher, 2012).


The data collection from experimental campaigns has been used to calibrate different types of radiative, convective, and turbulent parameterizations (Ciesielski et al., 2017; Feng et al., 2015; Moulin et al., 2018; Pujiana et al., 2018) and to explain meteorological phenomena at different weather and climate scales (Hagos et al., 2016; Mather, 2005; Rowe et al., 2019).



These field campaigns (GoAmazon and DYNAMO) helped to improve the understand of the cloud-radiation interaction
processes and the physical processes responsible for developing meteorological convective systems.

The great variability of cloud type and composition depends on the region where clouds are formed (Giangrande et al.,
2017). Clouds type and composition play a significant role in modulating global and regional climates, especially over South
America. It is a region with great convective activity, working as a great source of heat and humidity (Nunes et al., 2016;
Satyamurty et al., 2013; Yanai and Tomita, 1998), these factors and instability force the formation of different cloud types.

However, feedbacks from the interaction of radiation, cloud, and turbulence are not well understood. Thus, further studies on
these issues are essential to explain the physical processes in the atmosphere in the Amazon region, as the feedbacks
modulate the energy balance, which is essential for feeding local and remote precipitating meteorological systems.

Numerical modeling is another methodology used to try to understand better the feedbacks interactions of radiation, cloud
and turbulence. Numerical models have a hierarchy based on the degree of precision of their parameterizations and
simplifications in dynamic equations (Frassoni et al., 2018; Jeevanjee et al., 2017), such as: the Atmospheric General
Circulation Model (AGCM-3D), Single Colum Models (SCMs), Cloud Resolving Models (CRMs) and Large Eddy
Simulation (LES). Each model has a scale domain that atmosphere can be represented.

The LES and CRM models are high-resolution numerical models whose grid spacings are sufficiently refined to allow
explicit simulations of turbulent vortices and individual clouds over the entire lifecycle or part of it (Tao and Moncrieff,
2009).

LES models and CRMs have their foundations in the parallel development of two types of explicit cloud models, the first
(LES) is dedicated to the study of shallow cumulus clouds, smaller and with short duration and the second (CRMs) is
suitable for the study of deeper deep convective clouds with a longer life cycle (Frassoni et al., 2018; Tao and Moncrieff,
2009).

Several studies found in the literature show the efficiency of CRMs in simulating atmospheric phenomena (convective
systems) with high resolution (Bretherton and Blossey, 2017; Khairoutdinov and Randall, 2001, 2002, 2003; Mechem and
Giangrande, 2018). More specifically for the Amazon region, which is the region of interest in this article, there are
numerous studies carried out on various topics that use LES (Chamecki et al., 2020; Dias-Junior et al., 2015; Neves et al.,
2018) and CRM numerical models (Khairoutdinov and Randall, 2006). However, for this same region, no work was found in
the literature that uses CRM numerical modeling as forcing data from GoAmazon 2014/15. Therefore, the use of modeling
with CRMs in studies for the Amazon region, using more recent observational data can provide detailed information on the

evolution of the cloud life cycle and the interaction of radiation with cloudiness that is not possible to obtain with observed data.


Thus, obtaining high quality data is essential to help understand how physical processes related to the effects of cloud-radiation interaction influence the convective systems development in the tropical region. So, exploring the formation of these systems and rainfall variability with high-resolution modeling is important to obtain this information.

The main objective of this article is to understand the interactions between the dynamic and thermodynamic variables of the atmosphere and cloudiness. Thus, the results section was divided into three parts.  Initially, it will be investigated whether the SAM model consistently simulates the temporal series of precipitation rate, shortwave radiation flux, and integrated cloud fraction pattern. Secondly, it will verify the cloudiness impact on radiation fluxes and how the average atmospheric conditions of the previous day can influence the formation and evolution of cloudiness, using the large-scale variables
anomalies. Finally, the relationships between the cloud fraction and large-scale variables will be quantified, aiming to use this information to develop and adjust cloud fraction parameterizations.

Section 2 shows a brief description of the observed data, CRM and the experiments designs. Section 3 presents the results regarding the observational and complementary study using numerical modeling (CRM), followed by the main research
conclusions in section 4.

## 2 Data and Methods

### 2.1 Data and site descriptions

The observational data are from the field experiment called The Observations and Modeling of the Green Ocean Amazon (GoAmazon 2014/15). The GoAmazon experiment 2014/15, carried out from January 2014 to December 2015, had several
data collection points around Manaus/AM city in the central region of the Amazon basin (Fig. 1). The experiment´s focus was to study the interaction between vegetation-atmosphere, atmospheric chemistry, aerosol production, clouds, radiation and precipitation, with the aim of understanding and quantifying these interconnected processes (Macedo and Fisch, 2018; Machado et al., 2018; Martin et al., 2016).

For this article, it was used only data obtained at the point called ARM Site (Fig. 1), also referred as T3 (Martin et al., 2016), which is located in Manacapuru/AM city (3.2133°S; 60.5987°W). Data were collected through the Program Atmospheric Radiation Measurement (ARM) from sets of instruments called ARM Mobile Facility (AMF) and ARM Aerial Facility (AAF). These data, in general, are very important for the scientific community, because they allow doing detailed studies about the diurnal cycle evolution of clouds and the interaction with radiation fluxes (Giangrande et al., 2017). Additionally,



they are useful for numerical modeling studies, which can be used as initial condition data and large-scale forcing for the integration of CRM, LES and SCM types numerical models, for the development of physical parameterizations and this can be used as reference data for the evaluation of numerical simulations.

During the GoAmazon 2014/15 field campaign, two intensive data collection periods were carried out, called IOP1 (15 Feb

2014 - 25 Mar 2014) and IOP2 (01 Sep 2014 - 10 Oct 2014). These IOPs were defined aiming to better characterize the wet (IOP1) and dry (IOP2) periods of the Amazon region. Since this article aims to study the cloud-radiation interaction during the wet and dry seasons, the observed data used will be only those obtained during the IOPs.

The data collected during GoAmazon 2014/15 used in this article are related to the cloud microphysical and macrophysical

characteristics, downward longwave and shortwave radiation fluxes and large-scale variables (temperature, omega, relative humidity). In addition to the horizontal advection of temperature and humidity are from the Variational Analysis product - VARANAL (Tang et al., 2016). The list of observational data used and their references can be found in Table 1.

## 2.2 Model descriptions

The Cloud Resolving Model (CRM) used in this research was the System for Atmospheric Model (SAM), version SAM6.11.4 of July 2020 SAM (Khairoutdinov and Randall, 2003). The model was created from a Large Eddy Simulation (LES) model at the University of Oklahoma and later new parameterizations were implemented to transform it into a CRM. This model can be used in the LES version in simulations for shallow convection and like CRM mode for simulations in which there are deep convection clouds with a vertical velocity above 1m/s.


The SAM is a non-hydrostatic model with an anelastic dynamic core. It has five microphysics schemes, including single-moment microphysics, double-moment microphysics (Morrison et al., 2005) and Thompson microphysics (Thompson et al., 2008) and two radiation schemes, being CAM3 Radiation (Collins et al., 2006) and the RRTM (Iacono et al., 2008).
The surface fluxes can be prescribed or simulated using the coupled surface model called the Simplified Land Model (SLM)

(Lee and Khairoutdinov, 2015). SLM was developed for use in Cloud Resolving Model and it has an interactive vegetation layer on the ground, and supports 17 classes of soil types. In addition to using as input data sand, clay, moisture and soil temperature content profiles.

## 2.3 Design of evaluation experiments

For this paper, the SAM model was configured using the single-moment microphysics scheme, the CAM3 radiation scheme

and the surface fluxes were calculated using the SLM model. The data forcing large scale (LSF) and initial condition (SND) data were extracted from the VARANAL product (Tang et al. 2016). In total, 8 simulations with different horizontal



resolutions, in wich 4 configurations for the IOP1 period and the same number of simulations for the IOP2 period. Each simulation was integrated by the total period of each IOP (40 days) with 64 vertical levels and varying only in the horizontal domains.


For both IOPs were used the following horizontal domains: 82,944Km² (grid 144x144x64 with a horizontal resolution of 2000m), 20.736Km² (grid 144x144x64 with a horizontal resolution of 1000m), 5.184Km² (grid 144x144x64 with a horizontal resolution of 500m) and 82,944 Km² (576x576x64 grid with 500m horizontal resolution). Table 1 summarizes the simulations settings.


The results section is divided into three parts and structured as follows:

Subsection 3.1 evaluates the temporal evolution of the integrated cloud fraction, precipitation rate and downward shortwave flux for the wet (IOP1) and dry (IOP2) periods. Additionally, it was verified the impact of the different horizontal resolutions

used in the simulations with the SAM model.

Subsection 3.2 shows the diurnal cycle of some atmospheric variables for two specific days, one within the wet period (IOP1) and the other to the dry period (IOP2). The analyzed variables are associated with macrophysical characteristics of clouds (cloud type, integrated cloud fraction and cloud fraction profile), incident radiation fluxes at the surface (long and

short wave) and anomalies of large-scale variables (temperature, relative humidity and omega). For each day are presented observational variables and those obtained from simulations performed with the SAM model. This section aims to evaluate the behavior of radiation fluxes as a function of the presence of different types of clouds and how the average atmospheric conditions of the previous day can influence the cloudiness.

Finally, subsection 3.3 shows dispersion figures correlating the cloud fraction variable with radiation fluxes and anomalies of large-scale variables. This procedure was performed both for observational data (subsection 3.3.1) and for simulations with the SAM model (subsection 3.3.2). The main objective is to quantify the cloud fraction values in relation to the anomalies of the large-scale variables.

## 3 Results

### 3.1 Horizontal resolution sensitivity and validation of the SAM model

The first part of the results evaluates the impact of horizontal resolution in simulations performed with the SAM model. The simulations are performed for the GoAmazon 2014/15 experiment IOP1 and IOP2 periods that representation the wet and dry season, respectively. Four different horizontal resolutions configurations were tested (see Table 1) for each period (IOP1





and IOP2), totaling 8 numerical experiments. The temporal evolution of domain-average precipitation rate, cloud fraction
and shortwave radiation fluxes simulated by SAM are compared with observational data

The temporal precipitation rate evolution (Fig. 2a) is generally well represented in the SAM simulations. However, despite
consistently simulating the observed patterns, the SAM model underestimates the peaks of maximum intensity of the
precipitation rate during IOP1. On the other hand, the simulations reproduce the observed daily precipitation cycle with well-
defined maximum peaks and precipitation rates above 2.5 mmh$^{-1}$. In representing precipitation diurnal cycle, the SAM
model performance, has already been shown by other authors, however, for other regions of the planet (Blossey et al., 2007;
Khairoutdinov and Randall, 2003). The good performance of the SAM model is due to the data used as large-scale forcing
(in this VARANAL/GoAmazon 2014/15 study) to represent the dynamics in CRM and SCM-type models. These forcings
are produced using precipitation data from radar products and observational measurements (Tang et al., 2016).


The cloud fraction (Fig. 2b) simulated by the SAM model presents consistent results with the observed data. During the
IOP1 time series (wet season) is observed that the maximum precipitation peaks (Fig. 2a) are associated with the maximum
cloud fraction values and lower values of downward shortwave flux (Fig. 2c). These results are physically consistent, as the
presence of clouds, especially deep convection clouds, tends to generate large precipitation volumes and reduce the amount
of solar radiation transmitted to the earth's surface.

It is important to mention that the observed cloud fraction is a punctual and indirect measure of the cloudiness condition. The
GoAmazon observed cloud fraction ia retrieved from the observed downward longwave radiation (Dürr, 2004; Riihimaki et
al., 2019). For an illustration, the 61st day of Julian (2nd march, 2014) stands out, even though simulations and observations
agree with the absence of rain and low incidence of solar radiation, the simulations diverge to observation in terms of the
cloud fraction daily cycle. The observation data indicates a temporal evolution of the cloud fraction throughout the day with
values below 0.6, while the simulations show values practically constant and close to 1. GOES-13 satellite images (not
shown here) show multilayer cloud type throughout the day over the experimental site. In this sense, the cloud fraction used
as a reference is a proxy for the cloud information. Thus, the importance of using, as a complement to the observed data, the
results of the simulations of the SAM model.

Figure 3 is similar to Figure 2, but for the IOP2 period (dry season). The SAM model simulated lower precipitation rates for
IPO2 (Fig. 3a) than for the wet period (IOP1), consistently with the observed data. Precipitation rate peaks of values below
2.0 mmh-1 are also correctly simulated, just like in the wet season (IOP1). The cloud fraction (Fig. 3b) indicates a smaller
cloud amount in IOP2 (dry period) compared to IOP1 (wet period), both for observational data and for simulations.
Downward shortwave flux (Fig. 3c) is higher in the dry period due to the smaller amount of clouds than in the wet period.
However, observations show that the last days of IOP2 are untypically rainiest and with the highest precipitation rate for a





dry period. The time series feature is well represented by the simulations. The Julian day 277 (4 October, 2014) stands out, when there is a decrease in the amount of downward shortwave flux due to the high frequency of cloud cover (Fig. 3b) and
precipitation occurrence (Fig. 3a).

In general, the SAM model can adequately simulate the different precipitation patterns, cloud fraction and, shortwave radiation flux observed in the wet (IOP1) and dry (IOP2) periods for the Amazon region. These results are initially related to the large-scale forcings (VARANAL/GoAmazon 2014/15) used in the simulations. These forcings are generated from the
observed precipitation, and they determine the large-scale conditions of the atmospheric systems acting in the study region for the SAM model. Furthermore, the consistent results are related to the physical parameterization options used in the simulations.
Other authors (Blossey et al., 2007; Khairoutdinov and Randall, 2003) have already shown the SAM model's ability to simulate the variables observed in other regions of the planet. However, for the dry and wet seasons of the Amazon region,
studies are not found in the literature with the SAM model.

Figures 4a and 4b show the histograms of cloud fraction distribution in the Amazon region, for the wet (IOP1) and dry (IOP2) periods. The cloud fraction distribution is presented for the observed data and simulations of the SAM model, with different horizontal resolutions.


In the period of IOP1 (Fig. 4a), a peak of maximum cloud fraction values above 0.9 is observed. The cloud fraction distribution is consistent with the meteorological characteristics of this period, in which there is more presence of clouds due to local convection and the presence of large-scale systems that favor convection in the region, such as instability lines (Cohen et al., 1995), CCMs and the ITCZ. The SAM model adequately simulates the cloud distribution pattern for the wet
season (IOP1) in the Amazon. The different horizontal resolutions were similar between them, with only a notable difference for the maximum values of cloud fraction, but not significant. Regarding observation, the SAM model with all resolutions has fewer cases with cloud fractions below 0.1. However, these cases of low cloudiness (fraction <0.2) are observed for concise periods during IOP1 (Fig. 2). For other cloud fraction values, the model simulates reasonably well, following the observational data distribution.


In the dry period (Fig. 4b), the observed and simulated pattern of cloud fraction distribution is inverse to the wet period, with a smaller number of cases with cloud fraction above 0.4. This pattern is expected, knowing that convection in the dry period is generated mainly by local factors with insignificant large-scale influences. The SAM (dry period) model showed a deficiency in simulating cloud fraction values below 0.4. This is an expected feature since the cloud fraction
parameterizations, in general, are unable to simulate partially clear sky conditions with the presence of shallow clouds. The simulations in all different horizontal resolutions represent well the cloud fraction distribution.





From the general analysis of the simulations carried out with the SAM model, it can be stated that the different horizontal resolutions used in this work presented satisfactory results of the patterns compared to the observed data. Therefore, it was

decided that the average ensemble between the 4 defined resolutions will be used in the following sections.

At this stage, it is shown that the simulations with the SAM model for the periods of IOP1 and IOP2, which occurred during the GoAmazon 2014/15 experiment, are satisfactory when analyzing the time series of precipitation, cloud fraction, and shortwave radiative flux. In the next section of the work, a more detailed discussion is carried out for the case of two specific days, a typical day of the wet period and another of the dry period. It expected to evaluate the skill of the SAM model and

the physical consistency between the variables related to the cloud-radiation interaction.

### 3.2 Daily cycle of large-scale variables and radiation fluxes

In this section, the results are discussed for two distinct dates and with typical characteristics of the wet (21 February 2014 - IOP1) and dry (04 October 2014 - IOP2) periods. In addition, the days were chosen depending on the presence of the types of clouds, especially when the occurrence of high, medium and low clouds were well defined during the day. The main

objective is to evaluate the behavior of large-scale variables and radiation fluxes in relation to the presence of different types of clouds. It is important to verify the consistency between the variables of the observed data obtained with different methodologies and analyze whether the SAM model simulates with consistency and accuracy the interaction between clouds, large-scale variables and radiation fluxes.

Figures 5a and 5b show the diurnal cycle, from 21 February 2014, of the variables obtained from observational data and from simulations performed with the SAM model, respectively. The analyzed variables are cloud types, cloud fraction profile, downward longwave and shortwave flux, precipitation rate, and column integrated cloud fraction. Additionally, the figures show the anomalies in relation to the previous day's average for the temperature, omega and relative humidity profiles.


The observed data (Fig. 5a) show the evolution of cloud types during the 21 February 2014. In the early hours of the day, there are cirrus clouds at high levels and clouds associated with shallow convection at low levels. At approximately 06:00 local time, when heating by solar radiation starts, there is an evolution from shallow clouds to Congestus-type clouds and later to deep convection clouds (between 08:00 and 13:00 local time). After deep convection and precipitation, for the rest of

the day, high clouds and some shallow low clouds are observed.

For the shortwave radiation flux, a negative effect is observed mainly in the presence of Congestus and Deep clouds. These type clouds a decrease the incident radiation on the surface. However, the greatest negative effect of the shortwave radiation flux occurs in the presence of deep convection, this is due to some characteristics of this type of clouds, such as great optical





and geometric thickness, in addition to a high and cold top. These types of clouds reflect some shortwave radiation back into space, and they absorb incident solar radiation. The line (red) showing the radiative effect of clouds (difference between fluxes in the cloudy sky and clear sky condition), clearly indicates a decrease in the shortwave radiation flux of about 800 Wm-² observed in the presence of a cloud-related to deep convection (Deep).

Despite a very humid atmosphere, longwave radiation fluxes are also altered in the presence of different types of clouds in the atmosphere. Clouds with a low base and great vertical development, such as congested clouds and deep convection, present greater radiative forcing, increasing up to 40 Wm-² the long-wave radiation descending to the surface. On the other hand, high clouds and formed by ice crystals, such as Cirrustratus, have a forcing of 20 Wm², while Cirrus has no effect in terms of descending longwave radiation.


The integrated cloud fraction shows values close to 1, indicating that the sky was completely covered by clouds during the early morning hours until approximately 13:00 local time. During this period, the presence of clouds of types Shallow, Congestus and Deep is observed. In the early evening, at 18:00 local time, there is a reduction in the cloud fraction, which is associated with the presence of some Cirrus clouds.


Large-scale variables such as temperature, omega and relative humidity were analyzed, as they are of great importance in the representation of macrophysical characteristics in cloud formation and are also used to estimate cloudiness in atmospheric numerical models, based on cloud fraction parameterizations (Geoffroy et al., 2017).

These variables were analyzed in the form of anomalies in relation to the average of the previous day. The average characteristics of the previous day can indicate in the calculation of the anomaly of large-scale variables (temperature, omega and relative humidity) if there is the performance of a large-scale system or more intense environmental conditions that may favor the formation of different types of clouds.

In Figure 5a, from 00:00 to 09:00 local time, a favorable behavior is observed in the atmosphere for the generation of deep convection, where there is a tendency for a negative temperature anomaly at medium levels and positive at low levels, strong upward vertical movements throughout the atmospheric profile and large amounts of relative humidity in the middle and lower atmosphere. After this time, in fact, the presence of clouds is observed associated with deep convection. Precipitation occurs between 04:00 and 11:00 local time and it is associated with shallow, congestus and deep clouds. After the

precipitation, there is a warming of the atmosphere due to the release of latent heat, weak subsiding vertical movements and positive relative humidity anomaly.





Figure 5b shows the diurnal cycle of the profile simulated by the SAM of cloud fraction: long-wave and short-wave radiation incident on the surface, integrated cloud fraction, precipitation, and temperature, omega, and relative humidity in the average

from the day before.

The pattern of the cloud fraction profile simulated with the SAM model is similar to the distribution pattern of the cloud types obtained from the observational data (Fig. 5a), however, the cloud fraction of the SAM model is described in terms of intensity, where 1 represents a completely cloudy sky and 0 represents a clear sky. During the diurnal cycle, in the early

hours of the day (00:00 to 06:00 local time), as well as in the observed data, there is the presence of clouds at low and high levels. From 06:00 local time, the SAM model shows clouds in all layers of the atmosphere, indicating the evolution to deep convection; however, this pattern persists only until approximately 08:00 local time. The observed data indicate that there is the presence of clouds generated as a result of deep convection up to 13:00 local time. After 13:00 local time (between 13:00 and 00:00 local time), the SAM model simulated the presence of clouds at high and low levels, as well as showing the

observed data.

The surface downward fluxes of shortwave and longwave radiation simulated with SAM model shows a similar behavior to the observed data, both in terms of the daily cycle pattern and in the magnitude of the values. The longwave radiation, as well as observed data, shows an increase in incident flux to the surface in the presence of clouds. Clouds with a low and

warm base, emit a greater amount of longwave radiation towards the earth's surface. The temperature, omega and relative humidity anomalies obtained from the SAM model showed a similar pattern but with a higher anomaly intensities modulus compared to the observed data.

Figures 6a and 6b show the diurnal cycle of the same variables as in Figure 5a (observed) and 5b (SAM model), but for one

day of the dry season (4 October 2014 - IOP2).

The diurnal cycle of the types of clouds (Fig. 6a) shows that in the early hours of the day (between 00:00 and 06:00 local hours), there is the presence of high (Cirrus) and low (Shallow and Congestus) clouds and soon after, the evolution of shallow clouds for deep convection (between 07:00 and 14:00 local time) and at later times it is possible to observe the

presence of high clouds until the end of the day (between 14:00 and 23:00 local time). Both shortwave radiation fluxes and longwave fluxes show variations related to the different types of clouds present in the atmosphere. However, during the presence of deep convection, the SAM model is unable to simulate the attenuation of shortwave radiation. The physical properties and amount of liquid water and ice and other hydrometeors can affect the attenuation of shortwave radiation by clouds. These and other physical considerations are discussed in the next sections.




The integrated cloud fraction shows higher values (approximately 1) at times with the presence of high/low clouds and where occur deep convection. Only in the presence of high clouds, the cloud fraction values are low, around 0.1 and 0.2. As mentioned before, the observed integrated cloud fraction is estimated from the longwave radiation flux; this can lead to estimation errors, mainly in the estimation of the high cloud fraction. Due to a large moisture amount at low levels in the
Amazon region, the longwave radiation flux emitted by high clouds is poorly transmitted to the surface. This is due to the absorption of descending longwave radiation by water vapor at low levels, which re-emits towards the surface. Thus, longwave radiation descending the surface is a little sensitive to radiation emitted by high clouds and this may explain the low cloud fraction values observed in the presence of high clouds of the type Cirrus and CirruSt. In simulations, the cloud fraction tends to be higher.


The anomalies in relation to the previous day's average of temperature, omega and relative humidity well describe the atmospheric behavior for the generation of clouds present during the diurnal cycle. Before the deep convection, the atmosphere has negative temperature anomalies, the omega variable indicates that there are upward vertical movements (negative omega) and there is the availability of water vapor for condensation indicated by positive relative humidity
anomalies. Soon after the precipitation that occurs in the presence of clouds related to deep convection (between 06:00 and 14:00 local time) positive temperature anomalies are observed, possibly related to the release of latent heat.

Figure 6b shows the diurnal cycle of the variables simulated with the SAM model. The simulation of the cloud fraction profile was similar to the pattern of cloud types obtained from observational data. The simulated values of the cloud fraction
profile, ranging from 0 for clear sky and 1 for completely cloudy sky, were able to clearly indicate the presence of different types of clouds (high, low and deep). In general, the cloud fraction pattern simulated with the SAM model well represented the pattern of the types of clouds present during the observed diurnal cycle. The shortwave and longwave radiation fluxes are impacted by the presence of different types of clouds in the atmosphere, mainly deep convection. These variations in radiative fluxes are also observed in the reference data (Fig. 6a).


The simulated integrated cloud fraction shows a maximum (approximately 1.0) at times when there are clouds related to deep convection. In the presence of high clouds are observed cloud fraction values of approximately 0.5. This indicates the good performance of the SAM model in simulating variables related to clouds.

### 3.3 Relation of large-scale meteorological variables and the cloud fraction

The last two sections qualitatively show that the SAM model has an excellent performance in the simulations of variables associated with clouds and large scale, as well as showing a good description of the processes of cloud formation and interaction of radiation and clouds. This section shows the relationships between large-scale variables (temperature, omega and relative humidity), cloud fraction and radiation fluxes from scatter plots. The objective is to analyze how the variables





are correlated with each other, as well as to understand the interaction of radiation fluxes with the cloud fraction and to

quantify the cloud fraction values in relation to large-scale variables.

It was selected 7 days of the wet period (IOP1) and 5 days of the dry period (IOP2). The criteria for the choice of days was based on the presence of a well-defined cloud cycle over the chosen days, in which it was possible to verify, from the observational (MERGE-RADAR/cloud mask), the presence of high clouds (Cirrus), clouds related to shallow convection

(Shallow) and deep convection (Deep). In this Figures were used all chosen days of each period and for the atmospheric layer between 965 hPa and 90 hPa. For the large-scale variables were used only the values referring to the environmental conditions that are associated with cloud formation, the minimum values for temperature anomalies, minimum values for omega anomalies (when negative, they indicate upward vertical movement) and maximum values for humidity anomalies relative. For the observed data was used (Fig. 7 and Fig. 8) the variable fraction of cloud integrated in the atmosphere, while

for the simulations with the SAM model (Fig. 9, Fig. 10 and Fig. 11) was used the cloud fraction profile.

### 3.3.1 The relationships between radiation, dynamics and thermodynamics with the cloud fraction in the observed data from GoAmazon 2014/15

It was analyzed the scatter plots between the integrated cloud fraction and the longwave (Fig. 7a) and shortwave (Fig. 7b) radiation fluxes, temperature and relative humidity anomalies (Fig. 7c), minimum temperature anomaly and cloud fraction

(Fig. 7d), minimum omega anomaly and cloud fraction (Fig. 7e) and maximum relative humidity anomaly with cloud fraction (Fig. 7f), for the wet period (IOP1) using observed data from GoAmazon 2014/15.

The scatter plot of longwave radiation flux versus cloud fraction (Fig. 7a) shows a positive trend, that is, higher values of cloud fraction are associated with higher values of longwave radiation flux. These results indicate that the radiative effect of

clouds for longwave radiation increases with increasing cloud cover in the atmosphere. On the other hand, the highest values of cloud fraction are associated with a decrease in the downward shortwave radiation flux (Fig. 7b), mainly in the presence of Deep clouds (yellow dots). These clouds have high albedo values, reflecting much of the incident solar radiation on the planet back into space, and additionally attenuate descending solar radiation.

Figure 7c shows the scatter plots between minimum temperature and maximum relative humidity anomalies and indicates a negative trend. In the first quadrant of the figure, it is observed that high (Cirrus and Cirrostratus) and medium (Altostratus) clouds are associated with positive temperature and relative humidity anomalies. In the second quadrant, positive temperature anomalies and negative relative humidity anomalies are observed for shallow convection (Shallow) and Cirrus clouds. In the process of formation of these clouds, the phase changes from water vapor to liquid water and ice tend to warm

the atmosphere.





The fourth quadrant shows that negative temperature anomalies and positive relative humidity anomalies are mainly related to clouds of the deep convection type (Deep - yellow dots). Positive relative humidity anomalies are associated with humidity horizontal advection and negative temperature anomaly are associated with vertical advection that favors the rise of warm and humid air, consequently the descent of dry and cold air to lower levels of the atmosphere cools the environment.


The large-scale variables (temperature, omega and relative humidity) that are commonly used in cloud fraction parameterizations are shown in Figures 7d, 7e and 7f, respectively. The cloud fraction tends to increase with the negative temperature anomaly. The warm and humid air parcels of the Amazon region rise, expand and condense forming clouds, and consequently cooling the environment. Deep convection clouds (yellow dots) are observed when the temperature anomalies are between -1 and -2 K and in the cloud fraction range of 0.4 and 1. For shallow clouds (red dots), it is observed that positives temperature anomalies are related to negative relative humidity anomalies, being associated with the water vapor condensation process.


Figure 7e shows that the cloud fraction has a positive trend with a negative omega anomaly. Minimum negative omega anomaly values indicate strong upward vertical movements in the atmosphere, favoring cloud formation.
No significant relationship was found between cloud fraction and relative humidity anomalies (Fig. 7f). However, clouds related to deep convection (yellow dots) only appear related to positive relative humidity anomaly values, indicating that for the formation of this type of cloud, it is necessary that the atmosphere is wetter.


Figure 8 shows the scatter plot for the same variables as in Figure 7, but for the selected 5 days of the dry period (IOP2). The behavior of the variables in relation to cloud fraction is similar to that found in the wet period (IOP1), and briefly listed: i) increase (decrease) of longwave radiation (shortwave Fig. 8a and 8b) with the cloud fraction, ii) despite not finding a relationship between minimum temperature anomalies and maximum relative humidity values, deep convection clouds are found in the fourth quadrant and are associated with negative temperature anomalies and positive relative humidity anomalies (Fig. 8c); iii) minimum values of temperature and omega anomalies, as well as in the wet period, during the dry season shows a tendency to increase the cloud fraction (Fig. 8d and 8e), that is, the decrease in temperature and upward vertical movements are associated with greater presence of clouds, mainly of the Congestus (orange dots), Shallow (red dots) and Deep (yellow dots) types.



In the dry period, it was observed a smaller amount of clouds of types Cirrus (blue dots) and CirruSt (cobalt dots) associated with maximum values of cloud fraction (between 0.7 and 1.0) in relation to the wet period (Fig. 8). These clouds in the wet season are associated with moisture availability and the formation of an anvil in clouds associated with deep convection.



### 3.3.2 - The relationships between radiation, dynamics and thermodynamics with the fraction of clouds in simulations with SAM


The discussion about the relationships between large-scale variables, radiation fluxes and cloud fraction are carried out for the simulations of the SAM model. For this analysis, the same days chosen for the observed data are used (7 days for IOP1 and 5 days for IOP2).

Figure 9 shows the scatter plots between the cloud fraction with the downward longwave radiative flux (Fig. 9a), shortwave radiation flux (Fig. 9b), temperature and relative humidity anomalies (Fig. 9c). In addition to the cloud fraction with the minimum temperature anomalies (Fig. 9d), minimum omega anomalies (Fig. 9e) and maximum relative humidity anomaly (Fig. 9f) for the wet period (blue) and dry period (red).

The incident flux at the surface of longwave radiation increases as the cloud fraction increases, both for the wet and dry periods. In the wet period, it is observed that there are more occurrences of high cloud fraction values (between 0.8 and 0.9) due to the performance of large-scale systems, together with local convection. The relationship between downward longwave radiative flux and the integrated cloud fraction is complex, because often the radiation that reaches the surface does not represent clouds at higher levels of the atmosphere, but a good relationship was obtained between the variables.


For short wave radiation fluxes, there is a tendency to decrease in relation to the increase in cloud fraction, due to absorption and cloud albedo. The correlation between the variables is not ideal, due to the presence of a lot of diffuse radiation in the atmosphere, which prevents a direct correlation between the shortwave radiation and the vertically integrated cloud fraction. The trends between long-wave and short-wave radiation fluxes relative to cloud fractions are consistent with those found in

the observed data. Figure 9c shows the scatter plot of minimum temperature anomaly values and maximum relative humidity anomaly values. As in the observed data is identified a negative trend.

Figure 9d shows the minimum temperature anomaly in relation to cloud fraction. It is observed that there is a tendency to increase the cloud fraction in relation to more negative values of temperature anomaly. The same behavior is observed for

the negative omega anomalies (Fig. 9e). These trends are observed for both the wet and dry periods. The relative humidity anomaly (Fig. 9f) shows an inverse behavior in relation to temperature and omega, the higher relative humidity anomaly values are associated with higher cloud fraction values. The behavior of these variables with respect to cloud fraction is physically consistent. When there are clouds, there is a reduction in temperature and upward vertical movements and it is necessary the availability of water in the atmosphere. These results also agree with those presented in the observational

analysis.





Figure 10 shows the scatter plot between liquid water content and cloud fraction, both simulated with the SAM model and for all atmospheric levels. A clear trend of increasing cloud fraction is observed as the liquid water content increases. Higher values of cloud fraction and water content are observed in the wet period (between 0.4 and 0.7 of cloud fraction and 0.10 and

0.18 g/Kg of WLC). The good correlation for both the wet season (0.75) and the dry season (0.74) indicates that WLC can be used to estimate the cloud fraction from physical parameterizations used in weather and climate numerical models.

## 4 Conclusions

The results obtained in this article aimed to understand the interactions between the dynamic and thermodynamic variables of the atmosphere and cloudiness. For this, observational data were used from the GoAmazon 2014/15 field campaign. As a

complement to these data was used a high-resolution numerical model (CRM-SAM). All analyzes were performed for the wet period (IOP1) and for the dry period (IOP2).

The time series of precipitation rate, integrated cloud fraction and downward shortwave radiative flux, simulated using the SAM model (with different horizontal resolutions) were compared with observational data obtained during the 2014/15

GoAmazon field campaign. The SAM model, even underestimating the maximum peaks of the precipitation rate, satisfactorily simulated the patterns of this variable compared to observational data. The integrated cloud fraction and shortwave radiation flux also follow the same pattern of precipitation rate. Both in the simulations and in the observational data, the maximum precipitation peaks are associated with maximum values of cloud fraction and with a decrease in the downward shortwave radiative flux. This behavior of shortwave radiation is related to the strong albedo of clouds, especially

deep convection clouds. In both periods (wet and dry), the simulations were consistent and this can be confirmed by the cloud fraction distribution histogram, which showed a higher occurrence of maximum values of cloud fraction (from 0.9 to 1.0), in the wet period and higher occurrence of minimum values of cloud fraction (from 0.0 to 0.4), in the dry period. This pattern of maximum (wet) and minimum (dry) of the cloud fraction variable is due to the wet period being characterized by the performance of large-scale systems (Instability Lines, CCMs and ITCZ) together with local convection, causing a higher

occurrence of clouds, mainly of deep convection and generating more precipitation in this period compared to the dry period.

For the analyzes referring to the temporal series of the wet and dry periods, one day of each period was chosen for a more detailed analysis of the diurnal cycle of the interactions between radiation fluxes, dynamic and thermodynamic variables of the atmosphere, and cloudiness. The variables temperature, omega and relative humidity were analyzed as anomalies in

relation to the average of the previous day.

On both days, the cloud fraction profile simulated with the SAM model showed a pattern similar to the observed data. The behavior of the surface incident shortwave and longwave radiation fluxes are physically associated with the cloud patterns





found in the observed data and in the simulation with the SAM model. It was possible to observe in the simulated data that there is a negative (positive) effect on the surface incident short (long) wave radiation flux, due to the presence of different types of clouds and the shape of the cloud fraction vertical distribution. In general, the impact on shortwave and longwave radiation fluxes is mainly associated with the presence of shallow and deep convection clouds.

Anomalies in relation to the average of the previous day of the large-scale variables presented behaviors physically associated with cloud formation, evolution and dissipation. In the evolution of clouds associated with shallow to deep convection, it was observed that there were positive (negative) temperature anomalies at low (medium) levels, intense upward vertical movements and positive relative humidity anomalies at low levels. Warming of the atmosphere was also observed after the occurrence of precipitation, possibly associated with the release of latent heat. This behavior was found in the observed data and in the simulation with the SAM. Thus, it is concluded that the anomalies of large-scale variables can be optimal estimates for use in cloud fraction parameterizations in numerical weather and climate models.

Regarding the analysis of the time series and the analysis of two chosen days, it was possible to conclude that the variables obtained observationally with different methodologies showed physical consistency between them. The only point to be highlighted is related to the variable vertically integrated cloud fraction, which is estimated from the longwave radiation. This estimate, in general, showed a consistent behavior, however, as it is a variable estimated from the longwave radiation flux, it may present some inconsistency in cases of atmosphere with high moisture content, such as the Amazon region. In relation to the SAM model, a good performance was observed when compared to the observed data. It also showed physical consistency between the analyzed variables, which was to be expected, since it is a model that simulates an integrated system of the atmosphere, based on sophisticated physical parameterizations and a prescribed dynamic based on large-scale forcings. Thus, the variables simulated with the SAM model can be used in cases when the variables cannot be obtained observationally, for example, cloud fraction vertical profiles, water and ice content.

The results of the dispersion analyzes confirmed that the downward longwave and shortwave radiative fluxes show alterations associated with the presence of clouds in the atmosphere, mainly of deep convection. A positive trend was observed between the cloud fraction and the longwave radiation, with a higher cloud fraction and higher long radiation flux. The opposite happens with shortwave radiation flux (higher cloud fraction values are associated with smaller shortwave radiation flux values). This behavior was found in the observed data and in the simulations with the SAM, thus showing the importance of consistently simulating the cloud fraction for a good performance of the radiation scheme in numerical weather and climate models, considering that the radiation schemes use cloud fraction information to estimate radiation fluxes.



Different types of clouds are associated with different environmental conditions represented by anomalies of large-scale variables. As an example, high and medium clouds are associated with positive temperature and relative humidity anomalies. Clouds related to deep convection, which modify radiation fluxes are mainly associated with negative temperature anomalies and positive relative humidity anomalies. This analysis showed that large-scale variable anomalies can be used for cloud fraction estimation. The water content profile is another important variable that can also be used to estimate cloud fraction, as it showed a positive trend and good correlation with cloud fraction.

With this research it was possible to understand the importance of correctly simulating the cloud fraction in weather and climate numerical models. As well as proposing the use of large-scale variables anomaly and physical quantities associated with clouds (water content), as a way to estimate the cloud fraction in physical parameterizations associated with cloudiness.

*Data availability.* The data used are available in the ARM Climate Research Facility (GoAmazon 2014/15 - https://www.arm.gov/research/campaigns/amf2014goamazon)

*Author contributions.* LG, PK, SC idealized this study, LG manipulated the data and the model, DS contributed to the organization of this manuscript, and all authors participated in the discussions and contributed to the writing of this manuscript.

*Competing interests.* The authors declare that they have no conflict of interest.

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




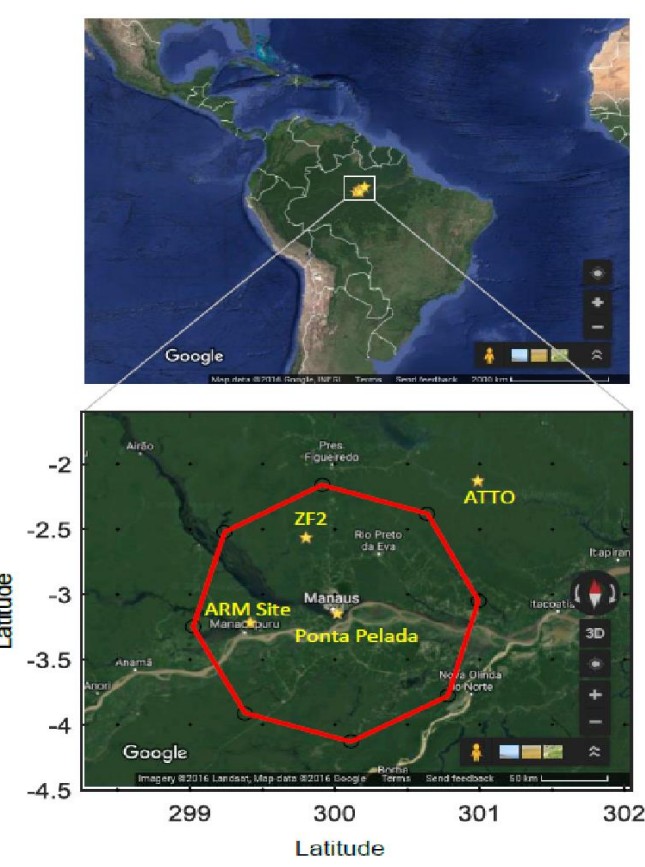


**Figure 1 - Data collection area of the GoAmazon 2014/15 Experiment and domain of VARANAL analyzes (red octagon). Images extracted from: Map data ©2016 Google, INEGI and Imagery ©2016 Landsat, Map data ©2016 Google.**

**Source: Tang et al. 2016**








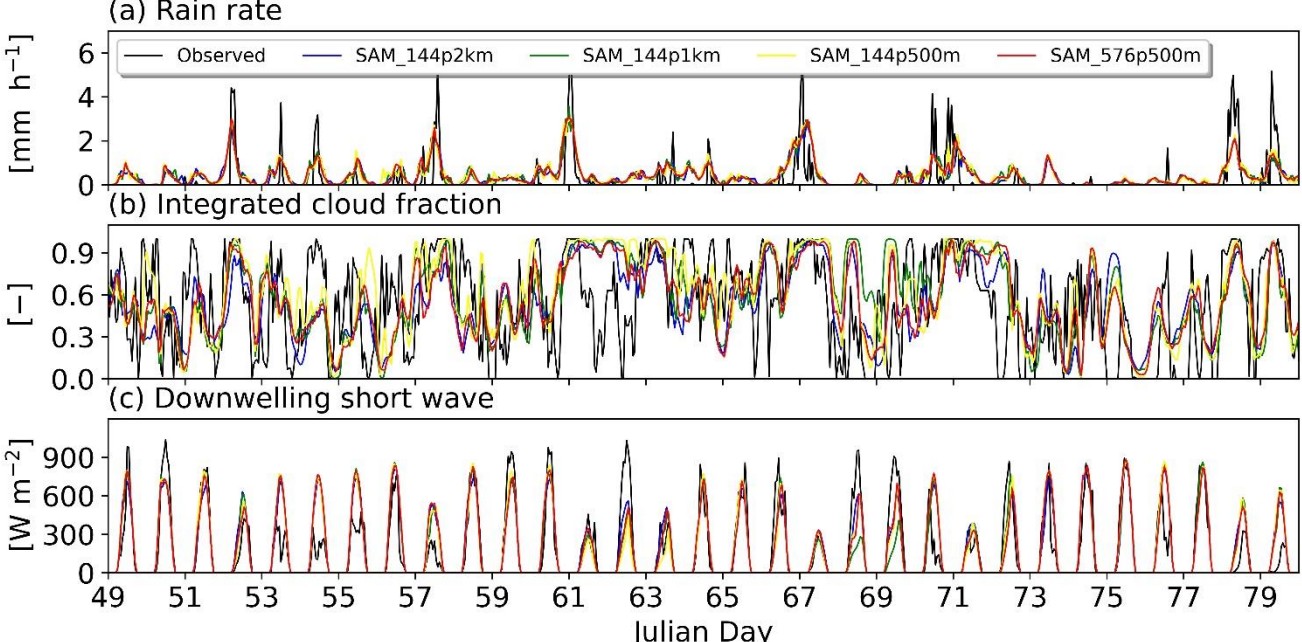

**Figure 2 - Time series of the domain averaged of the column integrated cloud fraction (ranging from 0 for clear-sky to 1 for overcast sky 1), precipitation rate (mm h⁻¹) and downward shortwave flux (W m⁻²) for the observed and the simulations with the SAM model for the IOP1 period (wet season). The black line represents the reference data (GoAmazon 2014/15) and the other colors represent the different horizontal resolutions used in the simulations (SAM).**






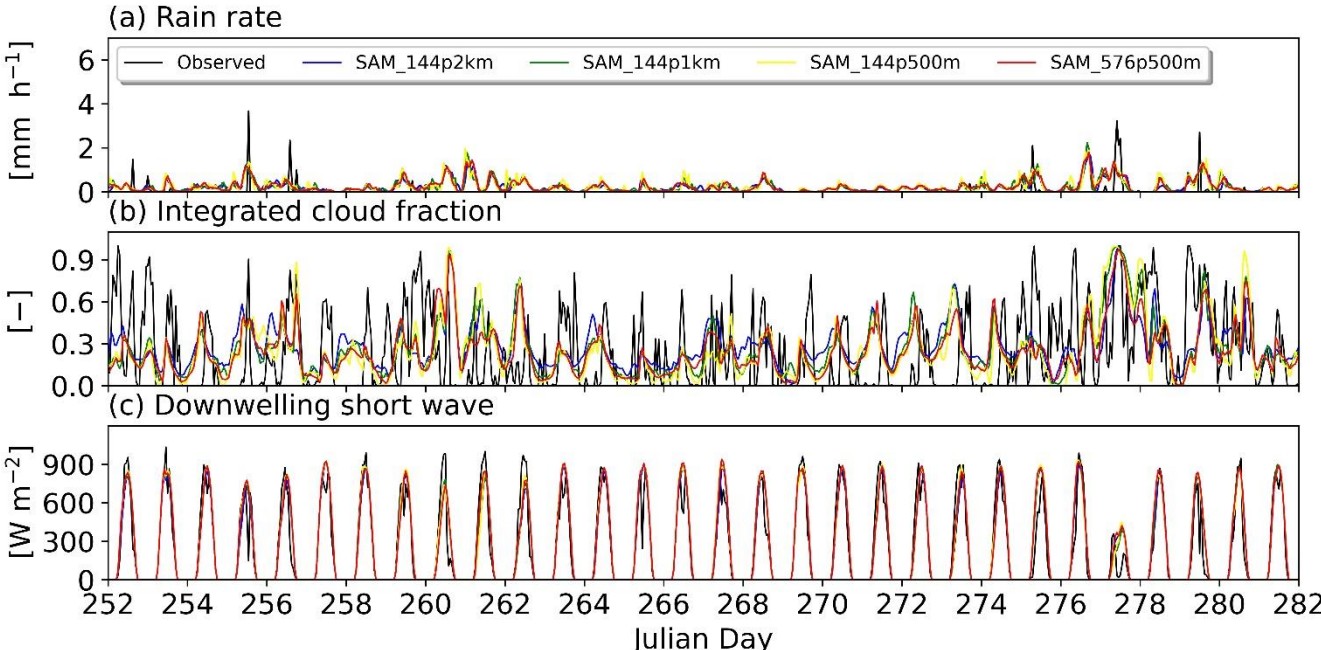


**Figure 3 - Time series of the domain averaged of the column integrated cloud fraction (ranging from 0 for clear-sky to 1 for overcast sky 1), precipitation rate (mm h$^{-1}$) and downward shortwave flux (W m$^{-2}$) for the observed and the simulations with the SAM model for the IOP2 period (dry season). The black line represents the reference data (GoAmazon 2014/15) and the other colors represent the different horizontal resolutions used in the simulations**

840                                                                                      **(SAM).**








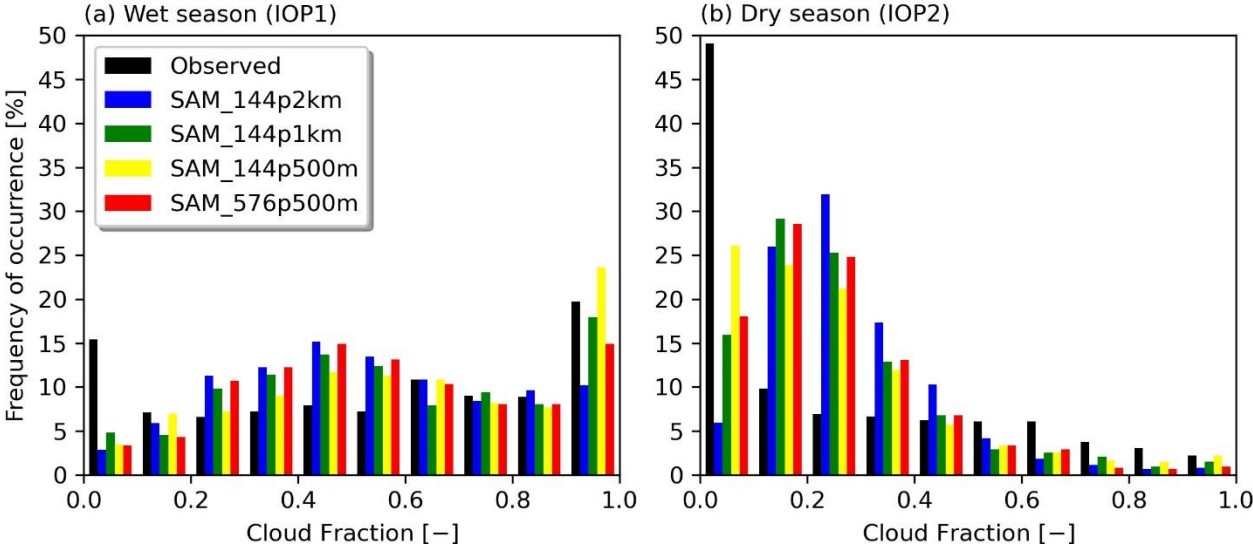

**Figure 4 - Histogram of the integrated cloud fraction distribution (clear-sky 0 and cloud-sky 1) for the observed and simulated data with the SAM model in different resolutions for the periods of IOP1 (a) and IOP2 (b).**







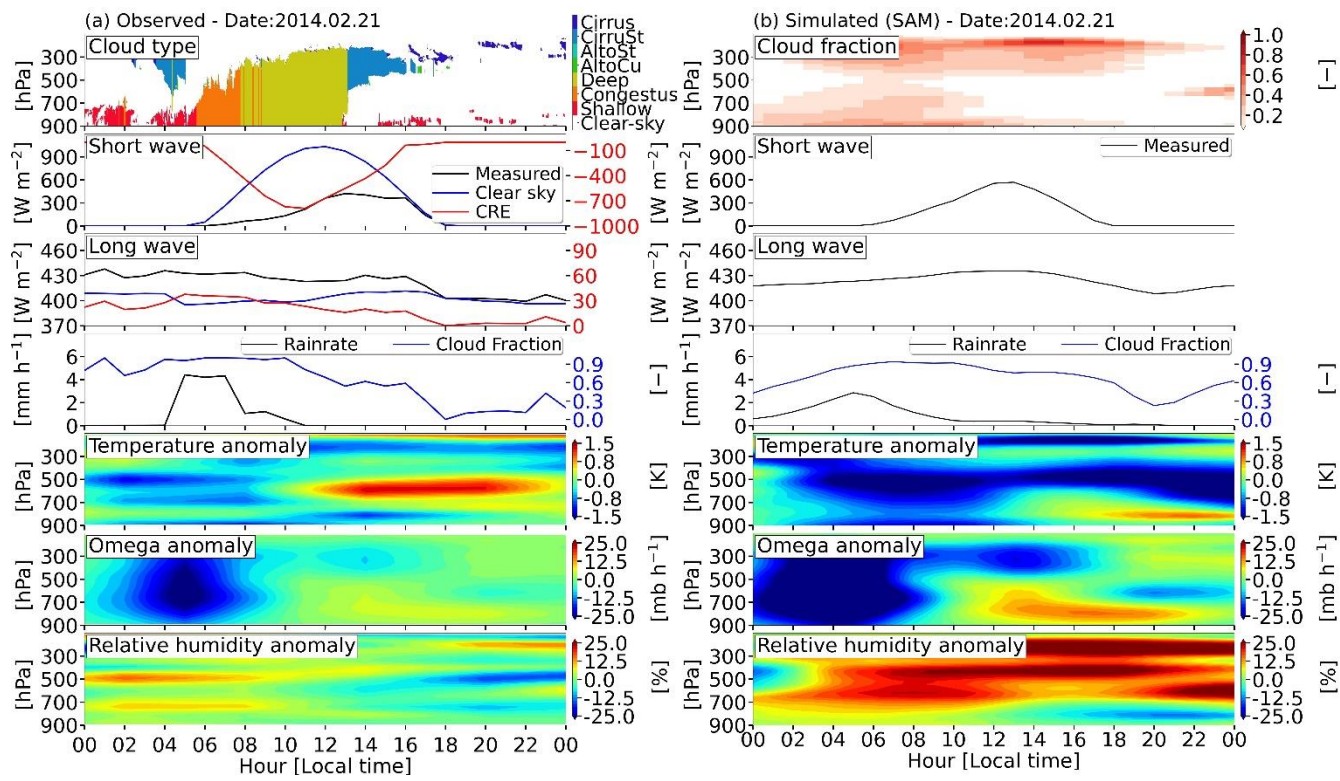

**Figure 5 - Daily cycle of cloud type, cloud fraction profile, longwave and shortwave radiation fluxes for cloud and clear-sky sky (W m⁻²), precipitation rate (mm h⁻¹), integrated cloud fraction and temperature (K), omega (mb h⁻¹) and relative humidity (%) anomaly from 2014.02.21 for the (a) observed data and (b) simulated with the SAM model.**








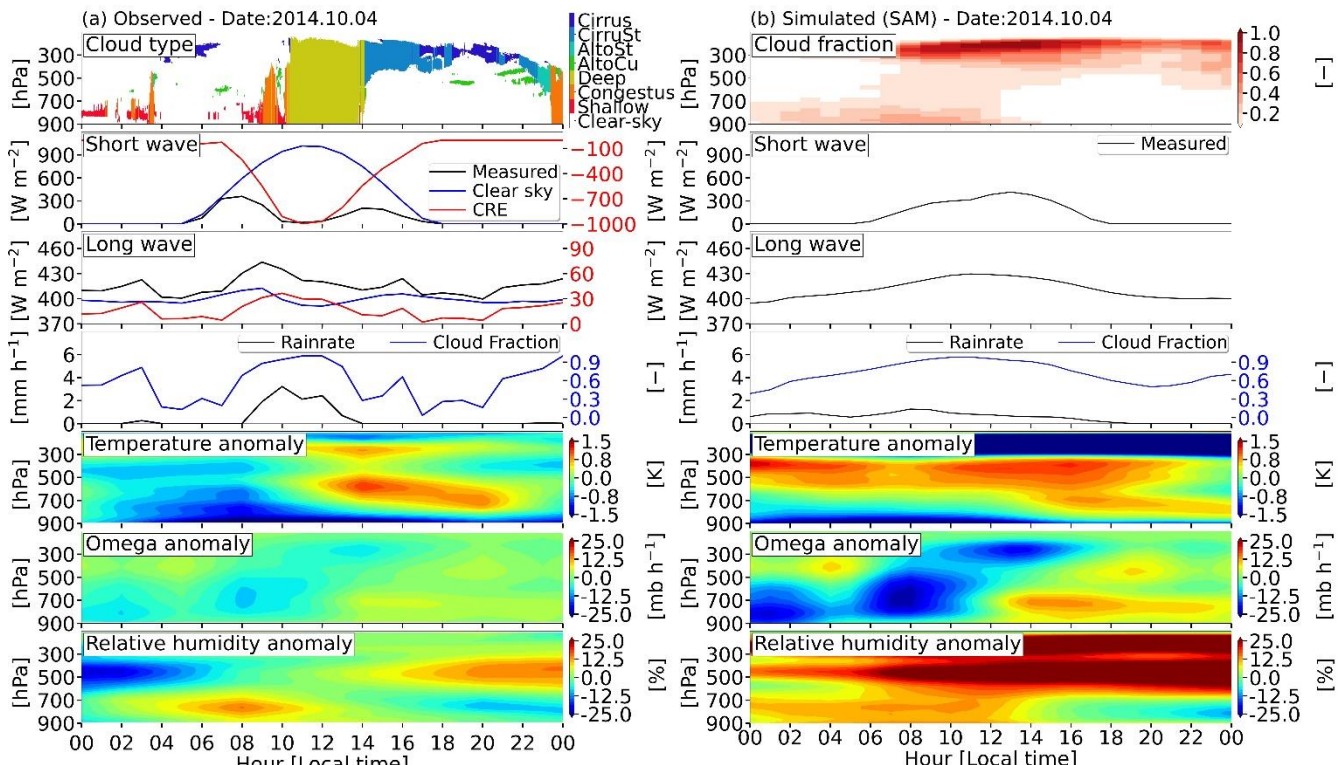

**Figure 6 - Daily cycle of cloud type, cloud fraction profile, longwave and shortwave radiation fluxes for cloud and clear-sky sky (W m⁻²), precipitation rate (mm h⁻¹), integrated cloud fraction and temperature (K), omega (mb h⁻¹) and relative humidity (%) anomaly from 2014.10.04 for the (a) observed data and (b) simulated with the SAM model.**









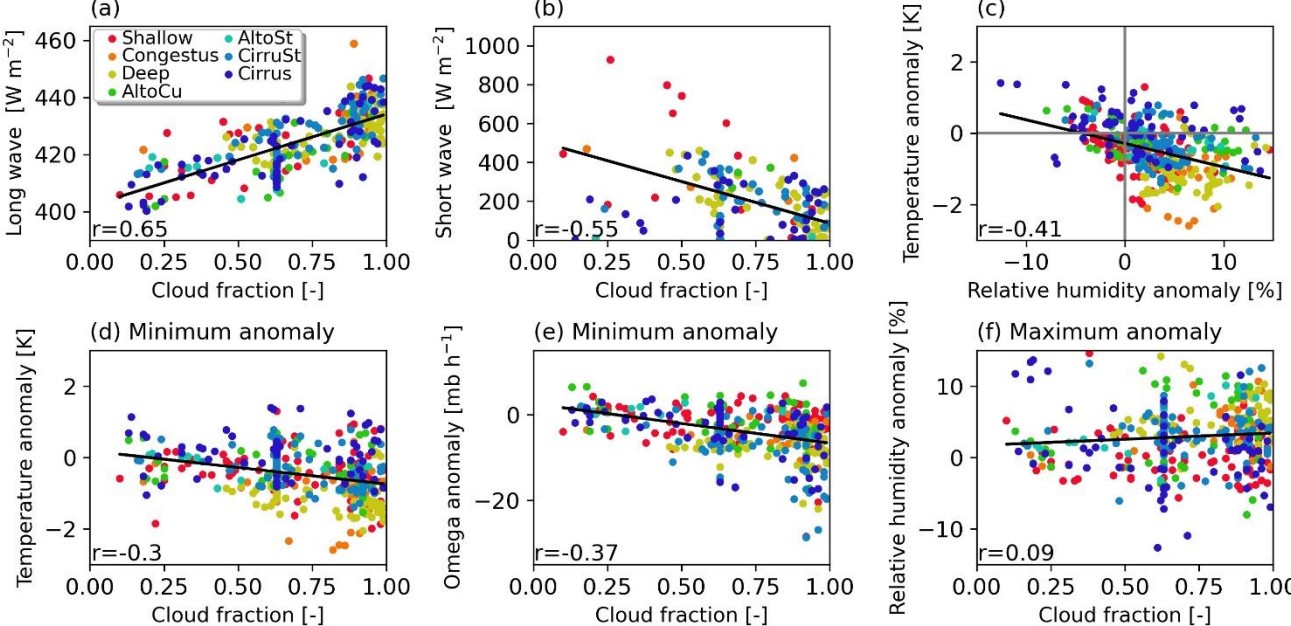

**Figure 7 - Scatter plot between observed cloud fraction and longwave radiation, shortwave radiation, precipitation rate, minimum values of temperature variance, minimum omega and maximum relative humidity for selected cases during the IOP1 (Wet season). Each color represents a cloud type, blue (Cirrus), cobalt (CirruSt), teal (AltoSt), green (AltoCu), yellow (Deep), orange (Congestus), red (Shallow) and white (clear-Sky).**









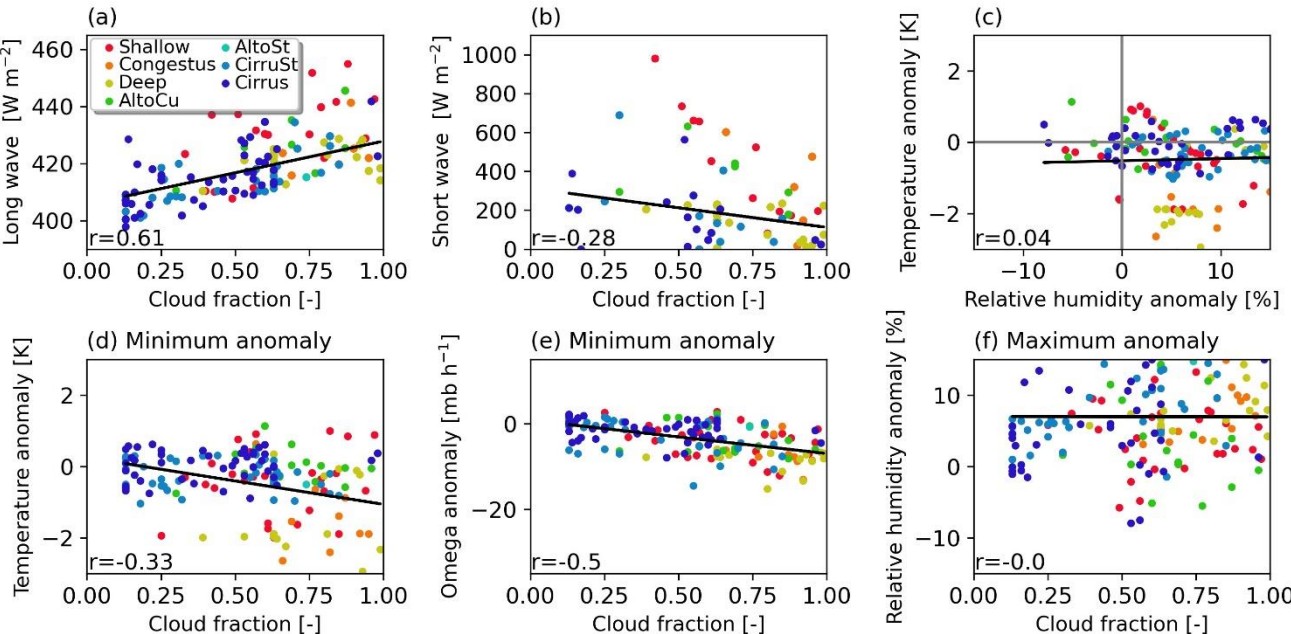

**Figure 8 - Scatter plot between observed cloud fraction and longwave radiation, shortwave radiation, precipitation rate, minimum values of temperature variance, minimum omega and maximum relative humidity for selected cases during the IOP2 (Dry season). Each color represents a cloud type, blue (Cirrus), cobalt (CirruSt), teal (AltoSt), green (AltoCu), yellow (Deep), orange (Congestus), red (Shallow) and white (clear-Sky).**








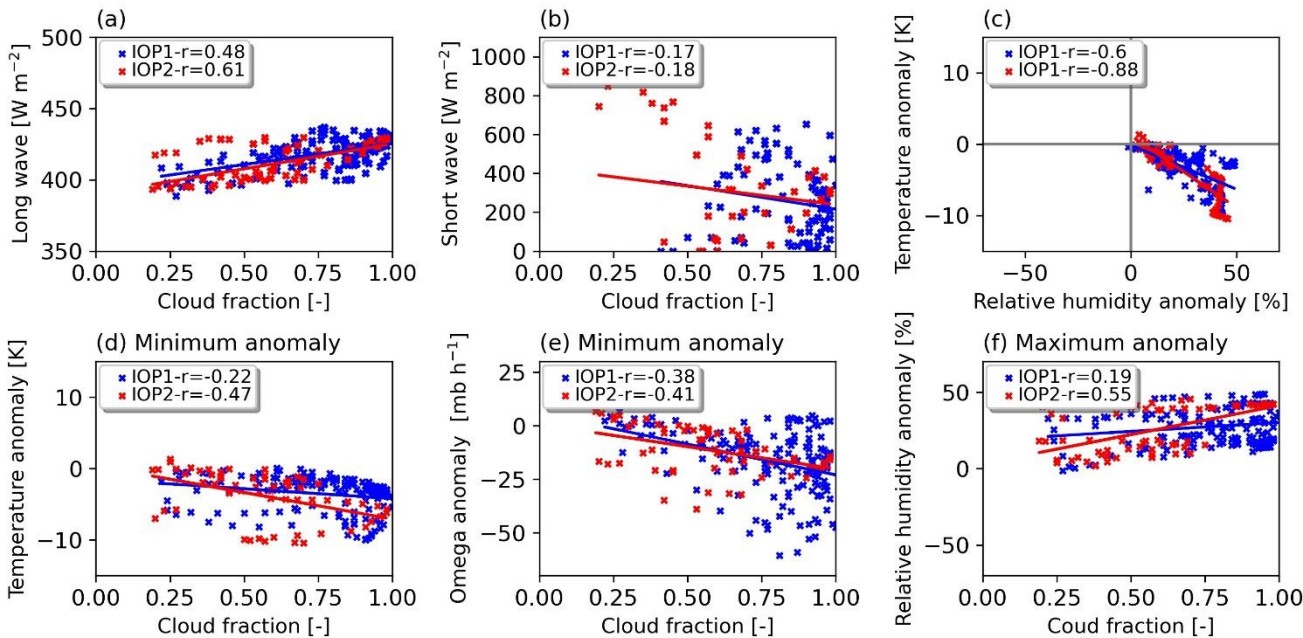

**Figure 9 - Scatter plots between the clouds fraction simulated with the SAM model and longwave radiation, shortwave radiation, temperature and relative humidity anomalies, minimum values of temperature anomalies, minimum omega anomalies and maximum relative humidity anomalies for selected cases during the IOP1 (Wet season - blue markers) and IOP2 (Dry season - red markers).**







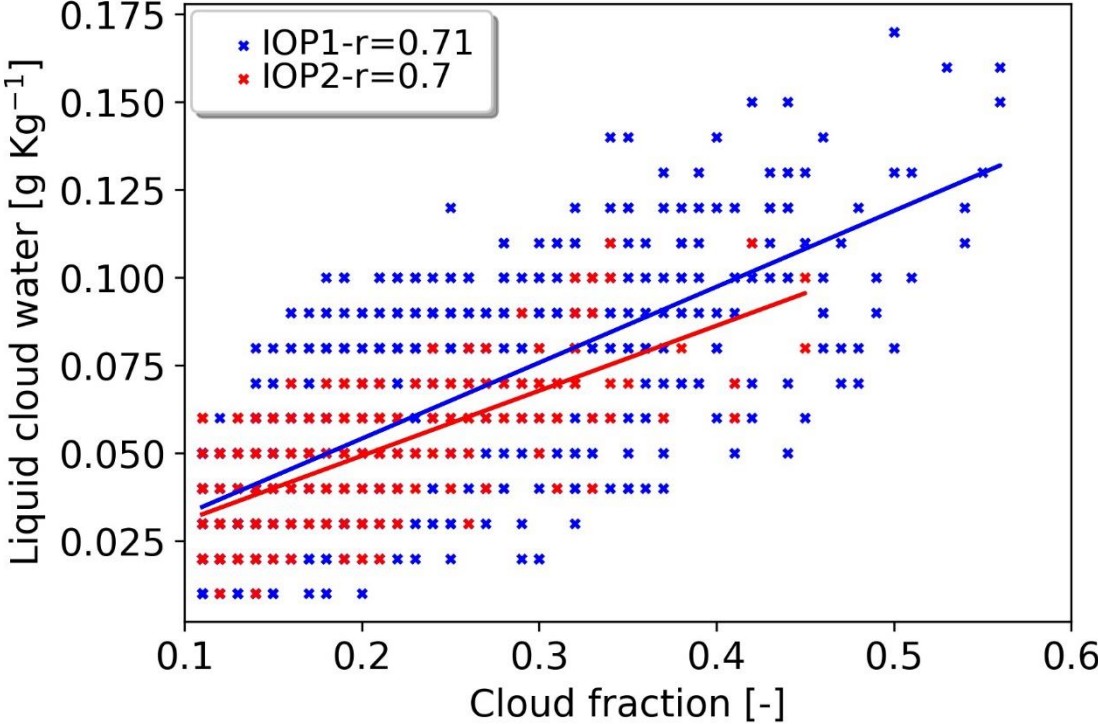

**Figure 10 - Scatter plot between the simulated cloud fraction and liquid cloud water content for the lower layer for cases selected during IOP1 (Wet season - blue markers) and IOP2 (Dry season - red markers).**






**Table 1 - Summary of observational data (GoAmazon 2014/15)**

| Product + | Variables | References |
|---|---|---|
| RADFLUXANAL | Downwelling longwave | Riihimaki et al. 2019 ARM, 2013 |
| | Downwelling longwave (clear-sky) | |
| | Downwelling shortwave | |
| | Downwelling shortwave (clear-sky) | |
| | Cloud fraction (estimated using longwave) | |
| Merged RWP-WACR-ARSCL | Cloud type merge | Feng et al. 2014; Giangrande et al. 2017 |
| | Rain rate | |
| VARANAL | Temperature | Tang et al. 2016 |
| | Omega | |
| | Specific humidity | |
| | U wind component | |
| | V wind component | |
| | Horizontal advection of temperature | |
| | Horizontal advection of specific humidity | |
| | Average surface pressure | |

+ According to GoAmazon 2014/15 nomenclature.






**Table 2 - Horizontal Resolution of simulations with SAM. Np represents the horizontal point numbers.**


| Run SNp$\Delta$x | Domain (x, y, z) | $\Delta$x, $\Delta$y (m) | $\Delta$t (s) | Number of Days |
|---|---|---|---|---|
| S144p2000 | 144x144x64 | 2000x2000 | 10 | 40 |
| S144p1000 | 144x144x64 | 1000x1000 | 10 | 40 |
| S144p500 | 144x144x64 | 500x500 | 10 | 40 |
| S576p500 | 576x576x64 | 500x500 | 10 | 40 |
