# Peer review of "Interaction between cloud-radiation, atmospheric dynamics and thermodynamics based on observational data from GoAmazon 2014/15 and a Cloud Resolving Model"

_Atmospheric Chemistry and Physics, 2021_

## Author Response (AR1)

Responses to referees are in blue color and the main changes in the manuscript are in red color.

**REFEREE COMMENTS 1**

**RC1:** This study examines observations of cloud cover, radiation, precipitation and atmospheric thermodynamic variables from the ARM site located in central Amazonian during GoAmazon and compares them with output from a CRM. The investigation looks for relationships between these variables in the observations and model outputs to see what can be learnt about the interaction of the clouds with their environment and their impact on radiation. The Amazon region provides an excellent environment in which to study the evolution of moist convection and how it relates to the large-scale environment. The use of CRMs is also well established to simulate deep convection and provide additional insight into convective cloud evolution. The authors evaluate various aspects of the CRM's performance including a thorough investigation of the sensitivity of CRM results to the horizontal resolution and show that the standard 2km set does a good job of simulating the temporal variability of clouds, precipitation and radiation although higher resolution better captures the distribution of cloud fraction. The study finds strong co-variations in cloud fraction and surface radiative fluxes at the surface and some correlations between cloud fraction, vertical motion, and column anomalies in temperature and relative humidity. Such relationships are to be expected given the nature of clouds, convection and radiation. In a general sense understanding these relationships better could aid the development and evaluation of cloud parameterizations in large-scale models.

**Response AC1:** We thank the reviewer for carefully reading our manuscript and providing very thoughtful comments and suggestions. We are glad that the reviewer highlighted the main results aspects of this study. Please find below a detailed response to each of the comments.

**RC1:** The analysis looks mostly at correlations between the fractional cover of different cloud types and the min/max anomalies of T and RH in the column based on day-to-day variations. This is interesting from an observational point of view in explaining the daily variations in cloud cover and precipitation but the limitation here is that there is only a loose physical connection between these anomalies and what determines the development of these convective clouds.

**Response AC1:** In this article, one of the objectives is to understand how the variation of large-scale variables (such as omega, T and RH), in relation to the average of the previous 24 hours, impacts the diagnosis of cloud fraction and radiation fields. These anomalies are produced by the physical processes (entrainment, dentrainment, updraft, downdraft, static energy, etc.) related to convective clouds, shallow, stratus, cirrus, etc. However, in numerical models, cloud fraction parameterizations are based on macrophysics variables (such as temperature, omega, relative humidity), and cloud microphysics variables (as liquid water and ice concentration) [Slingo, 1987; Sundqvist et al., 1989; Roeckner, et al. 1996; Tompkins, 2002; Gettelman et al., 2010; Bogenschutz et al., 2012; Machulskaya 2015; Dietlicher et al., 2019; Muench and Lohmann 2020]. Therefore, using information related to the convective cloud's development only helps define the cloud´s top and bottom of the cloud in the cloud fraction parameterization. The information from the convective clouds development of convective clouds may not contribute to improving the cloud fraction parameterizations currently used in numerical models. It is important to mention that cloud fraction and deep convection parameterization are independent algorithms. We are glad about this comment, we can better clarify these aspects associated with correlation analyses in the manuscript, and make the reader note that the analysed variables are based on those used in numerical model parameterizations, mainly in the cloud fraction parameterization.

RC1: The vertical profile of temperature and moisture and the resulting stability or instability (CAPE, CIN etc) is also a crucial factor that is missing from the analysis, along with broader constraints such as the large-scale convergence of moisture. This may be why the cloud fractions display a lot of scatter in their relationships to the column anomalies of T, RH and omega and relatively low correlation coefficients.

**Response AC1:** CAPE and CINE are used to analyze the life-cycle of deep convection and these variables are considered in the deep convection parameterizations. Notice that this study is focused on cloud cover parameterization, and not deep convection parameterization. Because of this, the article has a more specific interest in analyzing the relationships between the diurnal variability of large-scale variables (temperature, omega, relative humidity) and the cloud fraction. Due to the use of point data from the GoAmazon experiment, the hypothesis adopted is that the information on the development of deep convection is already associated with diurnal variability of large-scale variables, as well as large-scale moisture convergence. Regarding the low correlation coefficient values found between the cloud fractions and the column anomalies of T, RH and omega, it is necessary to mention that the data of cloud fractions, liquid water and ice from the GoAmazon experiment are a restricted data and with availability limited. Therefore, the informations used as cloud fractions, liquid water and ice are obtained in this work through simulations with CRMs.

**RC1:** Moreover, the relationships observed during these IOPs are unlikely to be generalizable as they assume a certain degree of convective instability and hence sensitivity to the T and RH anomalies.

**Response AC1:** The IOP1 and IOP2 experiments are used to analyze the dry and wet periods in the Amazon region. In the IOP1 (wet) condition, the large-scale systems that act on the region of the GoAmazon experiment are active in this period, contributing to the convective developments, while in the IOP2 (dry) period, the performance of large-scale systems is very reduced in this period, not favoring the development of convection. We also agree that the results obtained cannot be generalized, however, the analysis of these two periods (IOP1 and IOP2) statistically represents well the convective activity of the region of the GoAmazon experiment.

**RC1:** Perhaps there is more that could be gained from this general perspective but it is not obvious from the conclusions how the analysis presented so far could be taken forward to aid the evaluation and development of parameterizations in large-scale models.

**Response AC1:** The results of this article are part of the Brazilian Atmospheric Model (BAM) development project (Coelho, et al, 2021a, 2021b, 2021c, Guimarães, et al. 2021, Figueroa, et al. 2016). All information obtained through this work is being used to develop and improve the cloud fraction parameterization used in the BAM model. A second article is being prepared focused on describing the new cloud fraction parameterization and its validation. We also glad for this comment, we could include this perspective in the manuscript.

**RC1:** For these reasons I find it difficult to recommend this study for publication in ACP.

**Response AC1:** We hope that our answers for the reviewer may have clarified their doubts and some points that were probably not clear in the article. We intend to take the above discussion into account in the final version. The suggestions and comments from the reviewer significantly can contribute to improving the publication quality.

**RC1:** The study would need to show an increased understanding of the physical interactions involved or a clearer path towards improving the physics in models.

**Response AC1:** We can clarify and direct the conclusions to show how to use these results to improve the cloud fraction parameterization in the models.

**REFEREE COMMENTS 2**

**RC2:** The manuscript "Interaction between cloud-radiation, atmospheric dynamics and thermodynamics based on observational data from GoAmazon 2014/15 and a Cloud Resolving Model" general goal, as stated, is to understand the interactions between the dynamic and thermodynamic variables of the atmosphere and cloudiness in Central Amazonia.

For that, the authors used a set of observational data collected during GoAmazon IOP's (dry and wet seasons) and carried out a set of simulations using a Cloud Resolving Model considering different spatial resolutions.

The first results are focused on the comparison between observed and modeled atmospheric variables (cloud fraction, rain rate, radiative fluxes, temperature, relative humidity, vertical velocity) looking at daily and diurnal variability.

The authors concluded that the model consistently simulated the observations

For the second part of the results, also focusing on the comparison between observations and modeling, the authors explore the relationship between cloud fraction and the atmospheric variables (short and long wave radiation, temperature, relative humidity and vertical velocity and liquid water content).

The authors concluded that shallow and deep convection clouds have significant impact on radiation fluxes in the Amazon region during wet and dry period, and that memory of previous day large-scale features (based on temperatura, RH, and vertical velocity anomalies) have a good correlation with cloud fraction.

I would recommend authors to carry out a careful revision of the manuscript; it seems that several grammatical corrections are necessary.

**Response AC2**: We thank the reviewer for taking the time to read our manuscript. We are happy that the reviewer understood and indicated the main points of the work. For final review, a more careful assessment will be made regarding the manuscript's grammar. Below are the changes and answers to the questions and suggestions raised by the reviewer.

**INTRODUCTION**

**RC2:** The introduction and the problem contextualization are somehow dispersed, the authors mention several aspects related to the importance of clouds and their interaction with radiation, in some points they mention aspects of large-scale atmospheric dynamics, little talk about thermodynamic aspects, they mention the types of models, but again there is a lack of connection between the contents that points to an objective characterization of the problem to be studied.

**Response AC2**: Thanks for the comment. We have taken this comment on the introduction and reviewed all the manuscripts for improvement and clarification (pages 1-4, lines 25-106).

[revised manuscript text omitted]

**RC2**: As the authors suggest developing and adjusting the parameterizations related to the cloud cover fraction, it would be interesting to discuss what are the limitations that they want to target, and the aspects that the proposed study would help to improve.

**Response AC2:** Thanks for the comment. As requested, we improved the contextualization of the introduction and in it we indicated the importance of representing cloud cover in atmospheric models and the main limitation of cloud fraction parameterizations. As well, we indicate how the results of this manuscript can be used in future works to improve the cloud fraction parameterizations. The text below was added to the introduction (page 3, lines 92-97) to clarify the limitations and what we are proposing in relation to cloud fraction parameterization.

In the context of the problem of cloud representation in numerical models of weather and climate, the cloud fraction schemes are highlighted, which are mostly based on relative humidity thresholds, and some important parameters for cloud fraction diagnosis. These parameters are usually empirically calculated and the choice of these values can generate uncertainties in the representation of cloudiness (Park et al., 2016; Geoffroy et al., 2017). This work aims to obtain information on variables related to the cloud itself, such as water and ice content, and large-scale variables (temperature, omega and relative humidity) to understand the conditions for the formation, maintenance or dissipation of clouds.

**METHODS**

**RC2:** Little is said about the site, about the presence of the city of Manaus, the characteristics of the region, circulation pattern, among other relevant information to reinforce the importance of the site. GoAmazon included several sites, each site was designed to meet different characteristics within the context of the interaction between the city of Manaus and the Forest, it would be interesting if the authors could describe a little more about the ARM site in the context of GoAmazon.

**Response AC2**: Thansk for the comment. This part of the manuscript the following paragraphs were included on page 4, lines 110-118 and on page 4, lines 122-125 .

The Amazon region plays a very important role in modulating the global and regional climate, especially over South America, as it is a great source of heat and humidity for the development and maintenance of precipitating meteorological systems. Due to variation in the annual circulation pattern and thermodynamic structure, the region has defined wet and dry seasons (Carneiro and Fisch, 2020), with annual rainfall totals of approximately 2200 mm (Marengo et al., 2018). The rainfall characteristics of the region are defined by the presence of different systems and meteorological phenomena throughout the year, such as the Intertropical Convergence Zone, Squall Lines, Friagens, River Breeze and Penetration of Frontal Systems and Convection Organization (FISCH et al., 1998). Due to the action of different meteorological systems together with local convection, the region has different types of clouds (Giangrande et al., 2017) and the interaction of these clouds with radiation is the focus of different studies and field campaigns carried out in the region.

**RC2**: In the method topic, the variables used are barely contextualized in the dataset description, The authors need to specify the macro and microphysical data that they are referring to.

**Response AC2:** The cloud microphysical variables that we refer are water and ice content, the cloud macrophysical variables are cloud fraction and cloud type. This part of the text will be modified as shown below (page 5, lines 144-151).

The data generated during the GoAmazon 2014/15 that were used in this article are related to the macrophysical (cloud fraction and type of clouds) and microphysical (water content and ice content) characteristics of clouds, downward longwave and shortwave radiation fluxes for clear and cloudy sky conditions and large-scale variables (temperature, omega, relative humidity). For the simulations carried out with the SAM model, large-scale forcing (horizontal advection of temperature and humidity) and as initial condition (surface pressure and profiles of temperature, specific humidity and U and V components of the wind) data from the Variational Analysis product - VARANAL (Tang et al., 2016). The list of observational data used in this article and the references where the methodologies adopted for the collection (instruments) or estimation (products) of each data can be found is found in Table 1.

**RC2:** Try to maintain consistency in relation to the description of the objective of the study, in the methods topic it is understood that what is intended is an analysis of the cloud-radiation interaction, but in in the introduction the focus of the study is described as to understand the relationship between dynamics, thermodynamics and the cloud-radiation interaction.

**Response AC2**: Thanks for this comment. We have clarified it in the new version manuscript.

**RESULTS**

**RC2:** The first part of the results focuses on evaluating the performance of the different model resolutions in relation to observation. I think that a statistical analysis to summarize the performance of each resolution would be helpful.

And it seems that an analysis separating different atmospheric scenarios, especially in the wet period, might bring interesting results. For example, in Figure 4 one can see that all resolutions fail in relation to the frequency of cloud cover fraction close to zero in the wet period, but at the other extreme, cases with a coverage fraction closer to 1 there is a resolution that seems to perform better than the others.

**Response AC2:** Thanks for the comment. One of the goals of the work is to analyze the behavior of the cloud fraction simulated by the SAM in the dry (IOP2) and wet (IOP1) periods, which are characterized by different circulation and precipitation patterns. Therefore, all analyzes and statistics are calculated separately for these two periods.

In Figure 4 it can be seen that the pattern of the distribution of different classes of cloud fraction is well simulated by the model, however there is difficulty in representing the frequency of the different classes of cloud fraction. Therefore, this is a shortcoming of the cloud fraction schemes used in CRMs, which generally depend on the turbulent regime and microphysical processes. Any deficiencies in these processes produce an inaccurate cloud estimate. The SAM model consistently simulates the lowest cloud fraction values in both periods.

Indeed, the simulation with a horizontal resolution of 144p1km (green box) in Figure 4 is the closest to the observed data for maximum cloud fraction values (1). However, we can not define this resolution as the best configuration, just because it presents a good accuracy to simulate maximum cloud fraction values in IOP1, as it would not be representative for all cloud fraction classes.

On page 9, lines 277-287, a Table was inserted with the values of the correlation coefficient, bias and RMSE between the observed data and the simulations using the 4 different horizontal resolutions and the average between them (ensemble) for the variables precipitation, integrated cloud fraction and flux of incident shortwave radiation on the surface.

In Table 3, the correlation coefficient of precipitation and shortwave radiation between the observed and simulated data presents a good correlation, indicating that the variability of the observed data is well simulated by the model. However, the BIAS and RMSEs indicate that the data simulated by the SAM are overestimated in relation to the observation.

Statistical analysis of the cloud fraction does not show satisfactory values for the statistical indices, probably due to the methodology for calculating the cloud fraction obtained with observed data (Riihimaki et al., 2019) and how it is parameterized in the SAM model (Khairoutdinov and Randall, 2003), which produces cloud fraction values distinctly.

Thus, from the histogram of the distribution of the integrated cloud fraction (Figure 4) and the statistical analyzes (Table 3), it was not possible to define a better configuration of horizontal resolution to be used in the work, so it was decided to use the average among the 4 resolutions (ensemble) for the other analyses.

**Tabela 3 - Summary of the statistical analysis. The correlation coeficiente (r), BIAS and RMSE were calculated between the observed data and each simulation with different horizontal resolutions. For the wet (IOP1) e dry (IOP2) periods.**

| | PRP | | | CF | | | SW | | |
|---|---|---|---|---|---|---|---|---|---|
| **Simulations (IOP1)** | r | BIAS | RMSE | r | BIAS | RMSE | r | BIAS | RMSE |
| SAM_144p2km | 0,63 | 0,16 | 0,68 | 0,34 | 0,00 | 0,34 | 0,85 | 11,04 | 144,24 |
| SAM_144p1km | 0,62 | 0,16 | 0,69 | 0,33 | 0,03 | 0,36 | 0,81 | 8,56 | 160,72 |
| SAM_144p500m | 0,62 | 0,17 | 0,69 | 0,41 | 0,07 | 0,35 | 0,82 | 9,37 | 156,62 |
| SAM_576p500m | 0,65 | 0,17 | 0,67 | 0,36 | 0,02 | 0,34 | 0,83 | 13,04 | 152,05 |
| SAM_Ensemble | 0,64 | 0,17 | 0,68 | 0,38 | 0,03 | 0,34 | 0,83 | 10,50 | 151,12 |
| **Simulations (IOP2)** | **r** | **BIAS** | **RMSE** | **r** | **BIAS** | **RMSE** | **r** | **BIAS** | **RMSE** |
| SAM_144p2km | 0,24 | 0,19 | 0,39 | 0,17 | 0,05 | 0,30 | 0,94 | 7,79 | 106,16 |
| SAM_144p1km | 0,20 | 0,19 | 0,41 | 0,24 | 0,02 | 0,30 | 0,95 | 14,29 | 104,44 |
| SAM_144p500m | 0,24 | 0,20 | 0,42 | 0,27 | 0,01 | 0,30 | 0,95 | 18,24 | 106,44 |
| SAM_576p500m | 0,23 | 0,20 | 0,40 | 0,24 | 0,01 | 0,29 | 0,94 | 17,91 | 108,31 |
| SAM_Ensemble | 0,24 | 0,19 | 0,39 | 0,24 | 0,02 | 0,29 | 0,95 | 14,55 | 105,28 |

**RC2:** Regarding the two study cases, the analysis focused on two days seems to me limited in relation to the objective of extracting consistent and robust relationships between the atmospheric characteristics of the previous day and the properties of the clouds. The authors should evaluate a more robust alternative.

**Response AC2:** Thank you for your comments about the number of days chosen for the study.

For the analyzes in section 3.2, they were evaluated every day within the periods of IOP1 (wet season) and IOP2 (dry season), however, to show in this work only two characteristic days were chosen (1 day of each period), in the which presented a well-defined diurnal cycle of different types of clouds, considering that the objective of this section was to evaluate the behavior of large-scale variables (temperature, omega and relative humidity) and the radiation flux in relation to the presence of different types of clouds. Furthermore, the performance of the SAM model (in this case using the ensemble between the 4 horizontal resolutions) in simulating the interaction between clouds and radiation fluxes and large-scale variables was evaluated. The importance of knowing whether the SAM model consistently and accurately simulates the role in cloudiness is due to the need to use model variables that are not possible to obtain observationally, in the case of this study, the liquid water and ice content.

In section 3.3, where the relationship between the large-scale variables, radiation flux and water/ice content with the cloud fraction is studied, aiming to propose the use of these variables in cloud fraction parameterizations, a larger sample of days was used , being 7 days for the wet period and 5 days for the dry period. These days were chosen by the presence and evolution of different types of clouds during the diurnal cycle.

On page 13, lines 390-396 and page 16, line 517, the paragraphs were improved in order to clarify the choice of only two days for section 3.2 and indicating the use of more days in section 3.3 where the relationship between the variables is evaluated.

**RC2:** To achieve the objective of integrating modeling as an element to understand the interaction between dynamics, thermodynamics, clouds and radiation, it seems that the model needs to be further explored. The presented design and analysis of the model output consisted essentially of an evaluation against observation.

**Response AC2:** Thanks for the comment. To clarify that in this work the SAM model was used mainly to complement the observational analysis, a paragraph was added on page 6, lines 169-172.

In this work, the SAM model is used, mainly, as a complement to the observed data. For this, the consistency of the model is verified in simulating the interaction between cloud cover and radiation fluxes and large-scale variables (temperature, relative humidity and omega) in order to use variables that are not available in the GoAmazon 2014/15, as well as to understand the importance of correctly simulating the cloudiness pattern in numerical models

**RC1 and RC2:** The discussions of the relationships between cloud fraction and the remaining atmospheric variables, in general, were focused on expected features, which make it difficult to identify a clear contribution regarding the needed development and improvement in cloud parameterization stated in the goals of the paper.

**Response RC2:** Thanks for the comment. To clarify, we have amended the introduction to better contextualize the limitations related to cloud fraction parameterization and the purpose of this work. At the end of the conclusion (on pages 18-19, lines 584-594), two paragraphs were added that indicate the contribution of this study to improve the cloud fraction parameterizations.

[revised manuscript text omitted]